# Synthesizing artificial devices that redirect cellular information at will

Yuchen Liu†*, Jianfa Li†, Zhicong Chen†, Weiren Huang, Zhiming Cai*

Guangdong Key Laboratory of Systems Biology and Synthetic Biology for Urogenital Tumors, Shenzhen Second People's Hospital, The First Affiliated Hospital of Shenzhen University, Shenzhen, China

**Abstract** Natural signaling circuits could be rewired to reprogram cells with pre-determined procedures. However, it is difficult to link cellular signals at will. Here, we describe signal-connectors—a series of RNA devices—that connect one signal to another signal at the translational level. We use them to either repress or enhance the translation of target genes in response to signals. Application of these devices allows us to construct various logic gates and to incorporate feedback loops into gene networks. They have also been used to rewire a native signaling pathway and even to create novel pathways. Furthermore, logical AND gates based on these devices and integration of multiple signals have been used successfully for identification and redirection of the state of cancer cells. Eventually, the malignant phenotypes of cancers have been reversed by rewiring the oncogenic signaling from promoting to suppressing tumorigenesis. We provide a novel platform for redirecting cellular information.

DOI: https://doi.org/10.7554/eLife.31936.001

**\*For correspondence:**
liuyuchenmdcg@163.com (YL);
caizhiming2000@163.com (ZC)

†These authors contributed equally to this work

**Competing interests:** The authors declare that no competing interests exist.

## Introduction

A basic ability of living cells is to sense extracellular signals by translating them into changes in regulation of cell signaling genes. They use the natural signaling network to execute complex physiological functions, such as cell survival, behavior and identity. As an interdisciplinary branch of biology, genetic engineering has developed rapidly during recent years with the objective of reconstituting the signaling network of the cell for therapeutic and biotechnological applications. Genetic devices have been used to construct novel signaling circuits such as genetic switches (*Gardner et al., 2000*; *Green et al., 2014*), digital logic circuits (*Moon et al., 2012*; *Ausländer et al., 2012*; *Siuti et al., 2013*), rewired signaling pathways (*Kiel et al., 2010*; *Yuan et al., 2012*; *Flock et al., 2014*) and feedback loops (*Stricker et al., 2008*; *Prindle et al., 2014*). It proved easy to construct networks between synthetic genes using standardized building blocks (*Zhang and Jiang, 2010*; *Schreiber et al., 2016*). By controlling the gene expression, there are both positive and negative gene connections. The key difference between them is that in positive connection, the regulated gene is activated for expression, while in negative connection, the regulated gene is silenced.

The native intracellular communication can be rewired using genetic devices that block or redirect signals, but connecting native input-output signals at will remains a challenge. For example, the previously developed trans-acting ligand-responsive RNA regulators (*Bayer and Smolke, 2005*; *Win and Smolke, 2007*; *Ausländer et al., 2010*; *Beisel et al., 2011*; *Chang et al., 2012*) can be used to engineer novel connections by inhibiting native gene expression in response to extracellular molecules. The conversion of signals into specific cellular events has been accomplished via inducible or repressible antisense RNAs or miRNAs. However, they can only build negative gene connections and the antisense or RNAi-based regulation often exhibits relative low efficiency. Our group has developed a newly-engineered class of genetically encoded devices–'CRISPR signal conductors' (*Liu et al., 2016*)–that can sense and respond to cellular signals of interest and in turn activate/

**eLife digest** Cells respond to signals from their surrounding environment. External signals activate a sequence of events inside the cell that can change how it behaves. These events are often called signaling pathways and they typically change the cell's behavior by changing the activity of its genes.

A major objective of the field of genetic engineering is to customize or artificially create new signaling pathways to make cells behave in certain ways. The ability to control a cell's behavior is likely to have a major impact on human health and medicine. For instance, it may be possible to reprogram signaling events in cancer cells so that they die rather than grow rapidly.

Researchers are developing artificial genetic devices to manipulate signaling pathways. Molecules of ribonucleic acid (or RNA) are widely used to design such devices. In nature, RNA molecules are highly versatile: messenger RNA molecules carry genetic information in a form that can be translated into protein, while other RNA molecules fine-tune gene expression and perform a host of other roles. RNA is apt for artificial devices because it can be tailored to detect signals and convert this information into a predictable outcome, such as turning specific genes on or off.

In 2016, researchers constructed an RNA device to control the expression of genes in response to particular signals. However, this device was too large to deliver efficiently inside cells. Now, Liu, Li, Chen et al. – including some of the researchers involved the 2016 study – design smaller RNA devices to overcome this limitation. Each new device consists of two RNA components: one that recognizes the signal, and another that recognizes the messenger RNA of a target gene. Together the two components trigger the desired change in gene expression in response to a specific signal.

The devices were shown to have multiple uses such as making new connections in a signaling pathway and creating new signaling networks. Furthermore, Liu, Li, Chen et al. engineered one device such that it was able to specifically turn off genes in a particular signaling pathway that allows human bladder cancer cells to divide. By silencing these genes, the cancer cells were less able to grow.

These newly developed RNA devices should allow other researchers to customize cellular information and may have future therapeutic applications as well.

DOI: https://doi.org/10.7554/eLife.31936.002

repress transcription of specific endogenous genes through a CRISPR interference or activation mechanism. The advantage of these devices is to construct both negative and positive connections between various selected biomolecules and it is only limited by the availability of functional RNA aptamers. However, they require an additional transgene encoding a large protein (dCas9) which further increases the complexity of the system. From an application point of view, the use of a compact RNA-based device is likely to be much more compatible with the limitations of transgene delivery technology than the use of a rather large protein-coding construct. Regretfully, except for miRNAs and siRNAs, no other RNA-based mechanism has been adapted as a wide-spread tool for controlling native gene expression.

In this work, we describe the multiple uses of 'RNA-based signal connectors' in mammalian cells to modulate translation of mRNAs transcribed from the native genome and from provided plasmids. This new technology acts at the translational stage, apparently promoting or suppressing recruitment of ribosomes to the target mRNA. Without the requirement for an exogenous protein, these small artificial RNAs can establish both negative and positive linkages between input and output signals at will. The work described here is an improvement on the past design of 'signal conductors' and shows that they can be used in multiple different applications.

## Results

### Design and construction of RNA-based signal-connectors

Previous studies have demonstrated that insertion of an RNA aptamer into the 5' untranslated region (5'-UTR) of the messenger RNA (mRNA) can reduce the rate of translation initiation through blocking ribosome scanning in the presence of ligand (*Werstuck and Green, 1998*; *Blount and*

*Breaker, 2006*). We hypothesized that a designed external complementary sequence can be used to hybridize to the target mRNA and to guide RNA aptamers for trans-regulation of cellular mRNA translation when a specific signal is present. To test this, we engineered RNA devices which use the antisense domain (a 20 nt antisense RNA) to recognize the mRNA of interest and the previously developed aptamer domain to control translation (*Figure 1A*). We used the signal-connector to tether a translational activation domain to enhance translation, or the aptamer domain alone to repress the translation. In principle, these modular devices which we called 'signal-connectors', can be designed to control the translation of any target mRNA in response to a signal-molecule of interest (*Figure 1B*) and thus link desired endogenous signals (input) to specific cellular signals (output) (*Figure 1C*).

## Signal-connectors effectively repress target gene expression

To test whether this approach could cause efficient repression of translation initiation (*Figure 2A*) and elongation (*Figure 2B*), we designed signal-connectors complementary to 12 different regions of the mRNA sequence of the Renilla luciferase reporter gene, either binding to the 5'-UTR or to the coding sequence (*Figure 2C* and *Supplementary file 1*). Each of these signal-connectors contained two segments: a 20 nt antisense RNA sequence designed to be complementary to the targeted mRNA sequence and two theophylline aptamer copies (*Jenison et al., 1994*). Of these 12 constructs, 11 induced significant decreases in Renilla luciferase expression in the presence of 1000 µM theophylline when they were stably transfected into HEK293 cells expressing Renilla luciferase (*Figure 2D*). The levels of luciferase activity did not change substantially in cells harboring signal-connectors grown in the absence of theophylline. Nuclear and cytoplasmic fractionation analysis showed that these constructs mainly located in the cytoplasm (*Figure 2—figure supplement 1*).

In addition, the repression activity seemed to be inversely correlated with the target distance from the 5' cap of mRNA, perhaps indicating that the expression of a target gene could be repressed more effectively at the early stage of translation. We then observed that addition of theophylline inhibited luciferase activity in a dose-dependent fashion (*Figure 2E*). We speculated that the effects of the signal-connectors should also be affected by the valency of ligands recruited to each mRNA target. To test this possibility, we then introduced one or three theophylline aptamers to the 3' end of signal connectors, constructed stably transfected HEK293 cells, detected the luciferase expression level and compared their effects with those of the 2 × signal connectors (*Figure 2—*

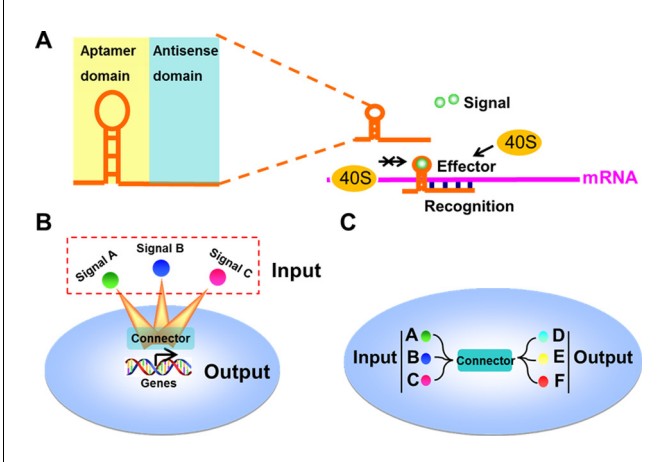

**Figure 1.** Design and construction of the signal-connectors for constructing linkages between signaling nodes. (**A**) Domain composition of a signal-connector. The device uses the antisense domain to recognize the mRNA of a target gene and the aptamer domain to respond to different signals to control the translation of the target gene. (**B**) Connectors that control the expression of cellular genes in response to specific exogenous signals can be engineered through this modular approach. (**C**) The signal-connectors can be used to direct the linkages between cellular inputs and outputs.
DOI: https://doi.org/10.7554/eLife.31936.003

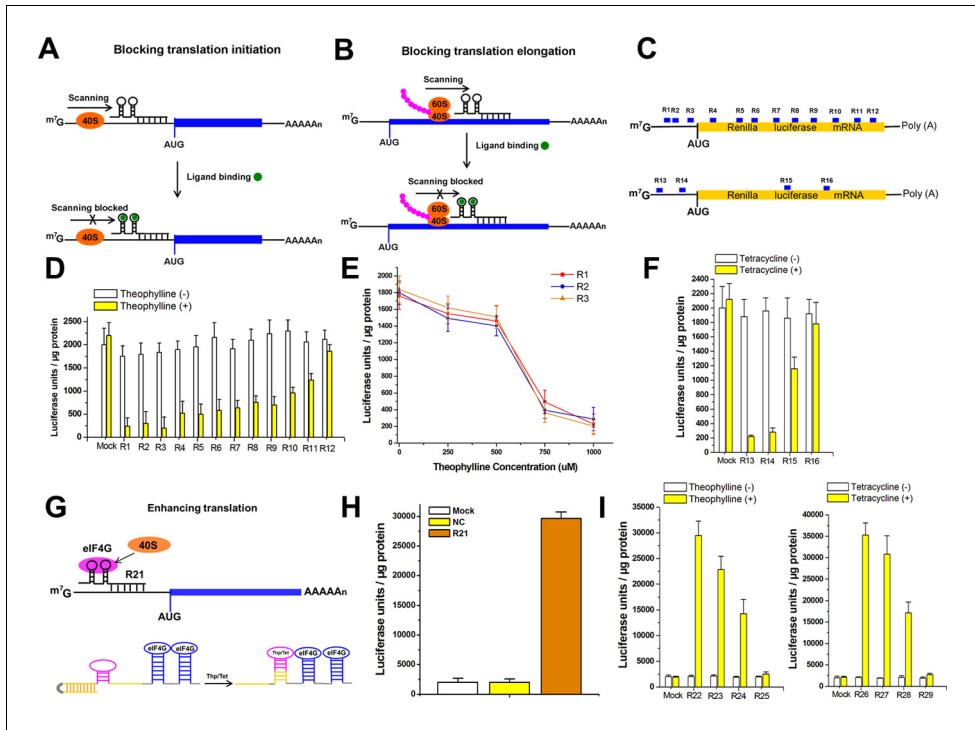

**Figure 2.** The signal-connectors effectively silence/activate gene expression. (**A**) Design of the signal-connector to block translation of the targeted gene. The device is designed to bind the 5′-UTR of the targeted mRNA. In the absence of the ligand, scanning of the 40S ribosome subunit proceeds until the AUG start codon is reached and translation is initiated. In the presence of the ligand, the ligand-aptamer complex disturbs ribosome scanning and blocks translation initiation. (**B**) When the binding occurs on the protein-coding region of target mRNA, it blocks ribosome and translation elongation. (**C**) Various signal-connectors were designed to target different regions of Renilla luciferase mRNA. (**D**) Renilla luciferase activity was suppressed by the signal-connectors only in the presence of 1 mM theophylline. Empty pGPU6/GFP/Puro vector as used as mock control. (**E**) Cells stably transfected with the signal-connectors (**R1, R2 and R3**) were grown in the presence of 0, 250, 500, 750, or 1000 µM theophylline. Addition of theophylline inhibited in vitro translation of the Renilla luciferase mRNA in a dose-dependent manner. (**F**) Suppression of Renilla luciferase in HEK293 cells by four different signal-connectors that bind to 100 µM tetracycline. (**G**) Design of the signal-connector for enhancing translation of a targeted gene. The activation of translation by a signal-connector is due to the activation of formation of initiation factor complexes involving eIF4G. In the absence of theophylline/tetracycline, the antisense domain is unable to bind to the mRNA of its target gene. In the presence of theophylline/tetracycline, the aptamer stem is formed and the antisense domain will bind to its target. (**H**) Renilla luciferase activity was increased by the signal-connector. (**I**) Addition of theophylline/tetracycline increased translation of the Renilla luciferase mRNA. Empty pGPU6/GFP/Puro vector as used as mock control. NC, negative control vector with two repeated elements not having targets in the human genome. Reported data are mean ± SD from at least three experiments.

DOI: https://doi.org/10.7554/eLife.31936.004

The following figure supplements are available for figure 2:

**Figure supplement 1.** Nuclear and cytoplasmic RNA fractionation analysis.

DOI: https://doi.org/10.7554/eLife.31936.005

**Figure supplement 2.** Comparing the repression effects of signal-connectors with different numbers of aptamers.

DOI: https://doi.org/10.7554/eLife.31936.006

**Figure supplement 3.** The in vitro translational repression activity induced by the ligand-aptamer complex.

DOI: https://doi.org/10.7554/eLife.31936.007

**Figure supplement 4.** The mathematical equation for gene repression induced by the signal-connector.

DOI: https://doi.org/10.7554/eLife.31936.008

**Figure supplement 5.** Suppression of VEGF protein expression in HEK293 cells by signal-connectors that bind to 100 µM tetracycline.

DOI: https://doi.org/10.7554/eLife.31936.009

**Figure supplement 6.** Relative expression level of VEGF mRNA.

*Figure 2 continued on next page*

*Figure 2 continued*

DOI: https://doi.org/10.7554/eLife.31936.010

**Figure supplement 7.** Comparing the activation effects of signal-connectors with different numbers of aptamers.

DOI: https://doi.org/10.7554/eLife.31936.011

**Figure supplement 8.** The in vitro translational activation activity induced by the ligand-aptamer complex.

DOI: https://doi.org/10.7554/eLife.31936.012

**Figure supplement 9.** The mathematical equation for gene activation induced by the signal-connector.

DOI: https://doi.org/10.7554/eLife.31936.013

**Figure supplement 10.** Addition of theophylline/tetracycline increased translation of the Renilla luciferase mRNA in a dose-dependent manner.

DOI: https://doi.org/10.7554/eLife.31936.014

*figure supplement 2*). Devices with three aptamers produced repression effects on luciferase expression that were little stronger relative to the analogous 2 × devices, perhaps due to saturation effects. Devices containing only one aptamer, however, only induced a very weak reduction in reporter gene expression.

We also performed an in vitro translation reaction using the macromolecular components (ribosomes, tRNAs, aminoacyl-tRNA synthetases, initiation, elongation and termination factors), purified ligand (theophylline), in vitro transcribed mRNA of Renilla luciferase, as well as in vitro transcribed RNA 'signal-connector' (R1 used in *Figure 2C*) or the negative controls. The in vitro data suggest that the observed silencing effects for signal-connectors were indeed induced by the ligand-aptamer complex (*Figure 2—figure supplement 3*), indicating a roadblock mechanism.

A simple mathematical model was then used to better understand the relationship between the various input parameters and the output (*Figure 2—figure supplement 4*). The equation described a dose- or concentration-effect relationship and a maximum effect, which are the key features of many biological phenomena. Based on the predictions of this equation and our observed results, we carried out gene knockdown experiments using designed 2 × signal connectors in the presence of sufficient amounts of ligand.

To demonstrate the modularity of this approach, we constructed several other signal-connectors to target the Renilla luciferase or the human vascular endothelial growth factor (VEGF) gene and replaced the theophylline aptamers with tetracycline aptamers (*Müller et al., 2006*) (*Supplementary file 2* and *Supplementary file 3*). The data on cells stably expressing these devices supported the modularity of the signal-connector to different aptamer domains (*Figure 2F* and *Figure 2—figure supplement 5*). As expected, these devices (R13 ~R15 and R17 ~R20) showed efficient silencing effects only in the presence of 100 µM tetracycline. Relative levels of VEGF mRNA did not change obviously between cells harboring the signal-connectors grown in the absence or presence of tetracycline, indicating that the signal-connectors function through translational inhibition rather than by affecting mRNA levels (*Figure 2—figure supplement 6*).

These results demonstrated that the signal-connectors could be used as gene switches to down-regulate the expression of a target gene.

## Signal-connectors effectively activate target gene expression

Using the RNA aptamers (*Miyakawa et al., 2006*) for eukaryotic translation initiation factor 4G (eIF4G), we then determined whether signal-connectors could also enhance translation of a target gene by promoting the formation of initiation factor complexes (*Figure 2G* and *Supplementary file 4*). eIF4G recruits the ribosome 40S subunit and activates mRNA translation (*Moore, 2005*). We chose the Renilla luciferase gene as the target gene and the results of luciferase reporter assay indicated that the specific signal-connector with two eIF4G aptamers induced a 15-fold increase in activity of luciferase protein relative to controls when they were stably transfected into HEK293 cells (*Figure 2H*). Elimination of one aptamer copy from the construct dramatically decreased the induced activation efficiency, whereas the fold change value increased minimally with the addition of another copy of aptamer (*Figure 2—figure supplement 7*). Nuclear and cytoplasmic fractionation analysis also showed that this construct mainly located in the cytoplasm (*Figure 2—figure supplement 1*). We also performed an in vitro translation reaction using the macromolecular components (ribosomes, tRNAs, aminoacyl-tRNA synthetases, initiation, elongation and termination factors except for

eIF4G), the purified eIF4G protein, in vitro transcribed uncapped mRNA of Renilla luciferase (the ORF encoding Rluc was placed downstream of a primary ORF), as well as in vitro transcribed RNA 'signal-connector' or the negative controls. The data suggest that the observed activating effects for signal-connectors were indeed induced by the eIF4G-aptamer complex (*Figure 2—figure supplement 8*), indicating a recruitment mechanism. We also constructed a simple mathematical model to clarify the relationship between the various input parameters and the output and the equation revealed that the relationship is nonlinear and saturable (*Figure 2—figure supplement 9*). Based on the predictions of the new equation for gene activation and the observed results, we carried out gene activation experiments with designed 2× signal connectors in the presence of sufficient amounts of ligand.

To achieve dynamic regulation of translation initiation, we used a combination of one aptamer recognizing theophylline or tetracycline and two aptamers recognizing eIF4G to regulate gene expression (*Supplementary file 5* and *Supplementary file 6*), in which the antisense domain was designed to be complementary to the stem sequence of the theophylline(or tetracycline)-binding aptamer (*Figure 2G*). Theophylline or tetracycline binding stabilizes the aptamer and leads to a conformational change that allows the antisense domain to interfere with the mRNA of the target gene. The results of the luciferase reporter assay on HEK293 cells stably expressing these devices indicated that addition of theophylline or tetracycline increased activity of luciferase (*Figure 2I*). We also observed a dose-dependent effect (*Figure 2—figure supplement 10*).

These results demonstrated that the signal-connectors could be used as gene switches to up-regulate the expression of a target gene.

## Construction of all the basic types of logic gates using the signal-connectors

In the construction of electronic circuits, logical calculations and digital systems can be practically implemented by using logic gates, including NOT, AND, NAND, OR, NOR, XOR and XNOR gates. Many aspects of information processing by biological cells are similar to signal integration of electronic circuits. We then asked the question whether the signal-connectors could be used to construct complex programmable logic gates and circuits. The excellent gene regulatory ability of the signal-connectors inspired us to construct various logic gates that produced output signals in response to multiple input signals through stably transfecting these devices (*Figure 3A*). We built all the basic types of two-input Boolean logic gates in HEK293 cells stably expressing the 5' capped or uncapped Renilla luciferase mRNA construct by using the aptamer recognizing exogenous theophylline or tetracycline signal. In the construct expressing uncapped Renilla luciferase mRNA, an open-reading frame (ORF) encoding Renilla luciferase was placed downstream of a primary ORF. The primary ORF contained a stop codon at the end.

First, we constructed two NOT gates, each of which produced an inverted version of the input at its output. As suggested in *Figure 2D and F*, the location of the antisense RNA target sequence along the mRNA was important for inhibition efficiency of the signal-connector. We used two devices (R1 and R13) that maximally suppressed translation of 5' capped Renilla luciferase mRNA in the presence of 1000 µM theophylline or 100 µM tetracycline to build these gates. As shown in *Figure 3B*, each NOT gate exhibited high luciferase output only in the absence of input signal.

We then constructed an AND gate that produced high output only if both input signals were high. As shown in *Figure 2—figure supplement 10* and *Figure 2I*, activation was also inversely correlated with the target distance from the 5' end of the mRNA. We used two signal-connectors (R25 and R29) that minimally activated translation of 5' uncapped Renilla luciferase mRNA in the presence of ligand to build this gate. As shown in *Figure 3C*, although introduction of each individual signal (1000 µM theophylline or 100 µM tetracycline) did not significantly stimulate expression of the target luciferase gene, the two devices acted synergistically to induce robust translational activation in the presence of both the input signals.

We also constructed a NAND gate that exhibited high output if any of the inputs were low. We used two signal-connectors (R12 and R16) that minimally suppressed translation of 5' capped Renilla luciferase mRNA in the presence of ligand to build this gate. As shown in *Figure 3D*, although introduction of either individual signal (1,000 µM theophylline or 100 µM tetracycline) did not significantly inhibit expression of the target luciferase gene, the two devices acted synergistically to induce robust translational repression in the presence of both input signals.

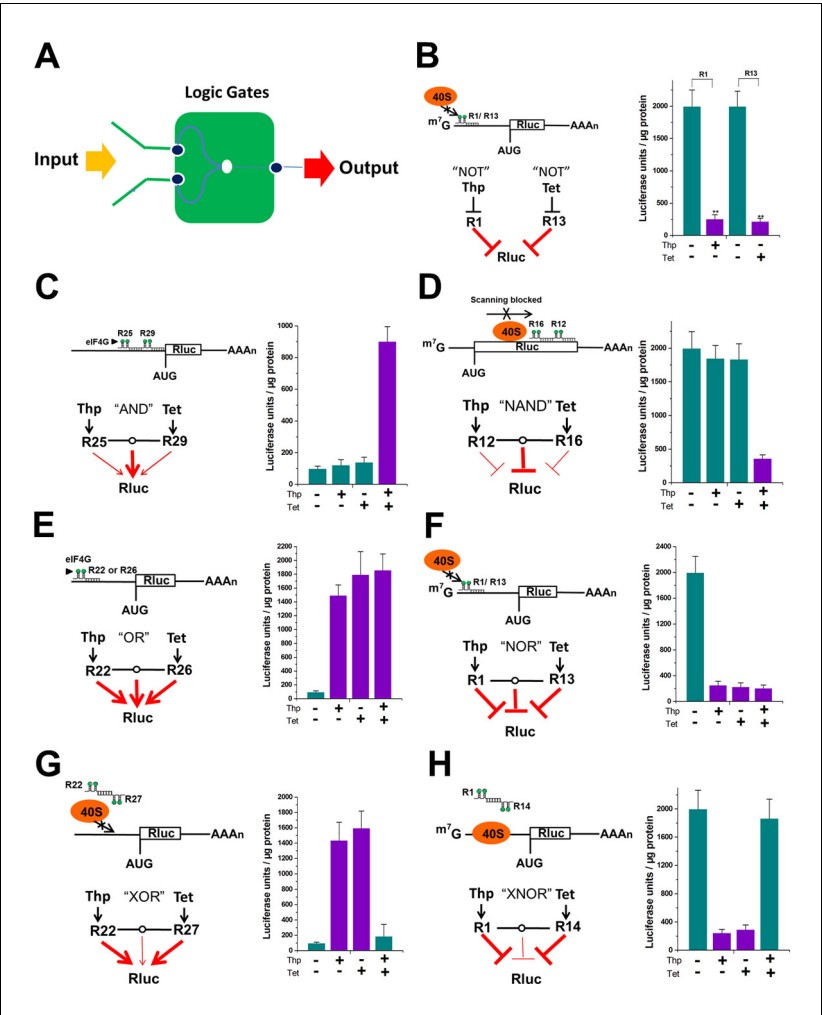

**Figure 3.** Logic gates based on signal integration were constructed by the signal-connectors. (**A**) In each logic gate, the signal-connector integrates the input signals theophylline and tetracycline and produces a luciferase output. Construction of gates that perform NOT (**B**), AND (**C**), NAND (**D**), OR (**E**), NOR (**F**), XOR (**G**) and XNOR (**H**) functions using the signal-connectors.

DOI: https://doi.org/10.7554/eLife.31936.015

Next, we constructed an OR gate that gave a high output if one or both of its inputs were high. In the building of this gate, we also used two signal-connectors (R22 and R26), each of which strongly activated translation of 5' uncapped Renilla luciferase mRNA in the presence of ligand. As shown in *Figure 3E*, the luciferase could be produced by either of the two signals (1000 μM theophylline and 100 μM tetracycline).

For our next test we constructed a NOR gate which was equivalent to an OR gate followed by a NOT gate. We used two devices (R1 and R13) that maximally inhibited translation of 5' capped Renilla luciferase mRNA in the presence of ligand to build this gate. Since introduction of individual signal significantly suppressed expression of the target gene, the luciferase could be produced only when both of the two signals (1000 μM theophylline and 100 μM tetracycline) were absent (*Figure 3F*).

We also constructed an XOR gate that exhibited high output if either, but not both, of its two inputs were high. We designed signal-connectors (R22 and R27) to target two different regions of the 5' uncapped Renilla luciferase mRNA, which were complementary in their RNA sequence. The results showed that each one of the devices strongly activated expression of luciferase in the presence of corresponding ligand. In contrast, introduction of both devices did not significantly activate

expression of the target luciferase gene due to the specific base pairing between their antisense domains (*Figure 3G*).

Finally, we constructed an XNOR gate that exhibited a low output if either, but not both, of its two inputs were high. Using similar design strategies, we used two devices (R1 and R14) that strongly repressed translation of 5' capped Renilla luciferase mRNA in the presence of corresponding ligand. The XNOR gate could have a high luciferase output when both 1000 μM theophylline and 100 μM tetracycline were present or absent (*Figure 3H*).

These results indicated that the signal-connectors could logically link input signals to a desired cellular output signal.

## Signal-connectors effectively rewire signaling pathways and create feedback loops

In eukaryotic cells, signaling proteins often activate transcription factors to initiate transcription of downstream genes. Because in theory the signal-connectors can link transcription factors to suppression of downstream gene translation, we set out to develop modifiers of a molecular network to

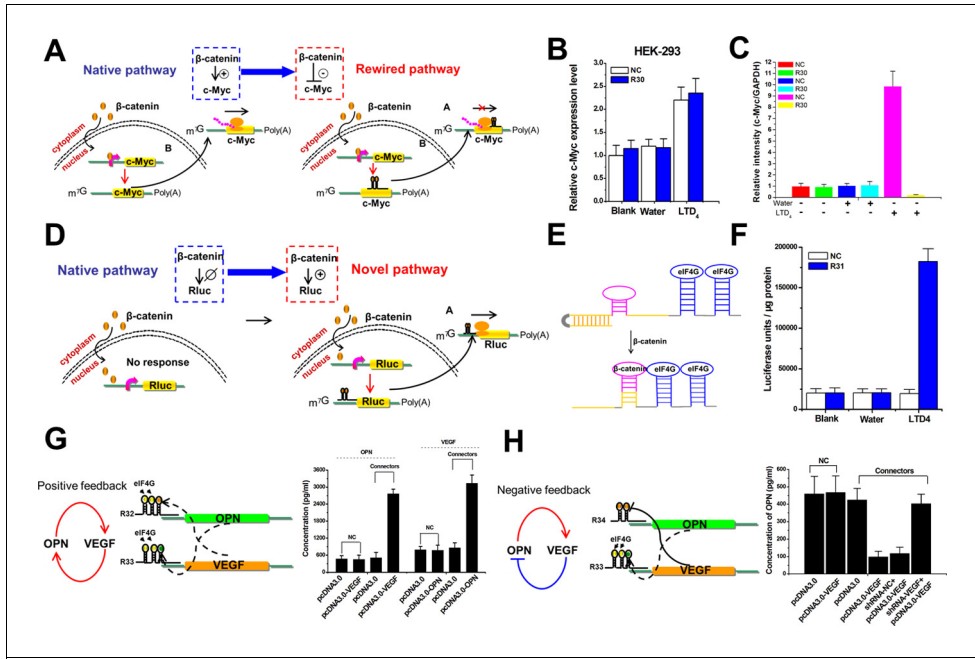

**Figure 4.** The signal-connectors effectively rewire and create signaling pathways and feedback loops. (**A**) Mechanisms of the signal-connectors designed to rewire the signaling pathway. (**B**) The relative expression levels of c-Myc mRNA were evaluated using real-time qPCR in HEK293 cells. The level of c-Myc mRNA was increased in cells that respond to $LTD_4$ stimulation. (**C**) Histogram of c-Myc. The values were normalized to GAPDH for each sample. The negative control was defined as 1.0. (**D**) Mechanisms of the signal-connectors designed to create the signaling pathway. (**E**) Design of the signal-connector that responds to β-catenin. This device frees the antisense region and targets the mRNA only in the presence of β-catenin. (**F**) The activity of Renilla luciferase was evaluated in HEK293 cells that respond to $LTD_4$ stimulation. (**G**) Designed models and experimental results illustrating the putative roles of the signal-connectors in constructing the OPN–VEGF positive feedback loop. (**H**) Designed models and experimental results illustrating the putative roles of the signal-connectors in constructing the OPN–VEGF negative feedback loop. NC, negative control vector with two repeated elements not having targets in the human genome. Reported values are presented as mean ± SD and the experiments were repeated at least three times.

DOI: https://doi.org/10.7554/eLife.31936.016

The following figure supplement is available for figure 4:

**Figure supplement 1.** Representative images of western blot analysis of c-Myc protein expression in cells transfected with the signal-connector or the negative control (NC).
DOI: https://doi.org/10.7554/eLife.31936.017

rewire the native signaling pathway (*Figure 4A*). β-catenin is a multifunctional protein and usually accumulates in the nucleus of cancer cells, where it activates the transcription of the oncogenic c-Myc gene (*He et al., 1998*). We synthesized a signal-connector containing β-catenin aptamers (*Culler et al., 2010*) to target the region within the 5'-UTR of c-Myc mRNA (*Supplementary file 7*). We investigated the effect of stimulating the β-catenin pathway with leukotriene D4 (LTD4) on the HEK-293 cells stably expressing either the signal-connector or the negative control. Both cell lines exhibited increased expression of c-Myc mRNA (*Figure 4B*), whereas the cells stably expressing the signal-connector showed a strong decrease in expression of c-Myc protein compared with the cells transfected with negative control (*Figure 4C* and *Figure 4—figure supplement 1*). These results demonstrated that our signal-connector could effectively rewire the signaling pathway by establishing a negative connection between the transcription factor and the mRNA of a downstream gene.

We also tested whether the signal-connector could create a novel signaling pathway by linking a regulatory factor to the activation of translation of a selected downstream gene (*Figure 4D*). The specific signal-connector used one aptamer domain to recognize β-catenin signal and the other two aptamer domains to form the initiation factor complexes. In the absence of β-catenin signal, the anti-sense domain was sequestered by the stem of the β-catenin aptamer. In the presence of β-catenin signal, this signal-connector could interact with the target Renilla luciferase mRNA (*Figure 4E* and *Supplementary file 8*). The effect of leukotriene D4 (LTD4) was investigated by stably transfecting HEK-293 cells with either the signal-connector or the negative control. The results of luciferase assay indicated that the activity of Renilla luciferase in cells expressing the signal-connector was obviously elevated by LTD treatment (*Figure 4F*), while its activity was not affected by LTD in cells expressing the negative control. These results demonstrated that β-catenin signal could effectively activate the expression of Renilla luciferase with the help of the designed signal-connector.

Next, we tested the ability of the signal-connectors to incorporate feedback loops into the gene–gene interaction networks. Positive feedback loops can amplify the cellular signal received from the sender and move a system away from its initial state. We engineered a feedback loop using the signal-connectors in which osteopontin (OPN) and VEGF were each other's activators (*Figure 4G*). OPN and VEGF are secreted proteins with cytokine properties and regulate cell motility and angiogenesis (*Ferrara et al., 2003*; *Lyle et al., 2014*) and their RNA aptamers were reported in previous literatures (*Ng et al., 2006*; *Mi et al., 2009*). We inserted two copies of eIF4G aptamer into the 3'end of OPN or VEGF riboswitch to construct signal-connector recognizing VEGF or OPN. The results of our over-expression experiments revealed that OPN and VEGF were mutually independently operated in bladder cancer T24 cells stably transfected with the negative control device. We transfected the plasmids over-expressing OPN or VEGF into the T24 cells expressing the signal-connectors (*Supplementary file 9*, *10* and *11*) and found that expression of the corresponding plasmid effectively increased the level of the regulated gene (*Figure 4G*). We also investigated whether the signal-connectors could be used to construct negative feedback loops between OPN and VEGF (*Figure 4H*), which could make the system more stable. Because OPN could induce the expression of VEGF via the signal-connector, we only needed to prove that VEGF could also inhibit the expression of OPN through a similar approach. We inserted two copies of VEGF aptamer into the 3'end of antisense RNA recognizing OPN to construct the signal-connector. Using T24 cells stably expressing this constructed signal-connector, we showed that transient expression of VEGF could decrease the concentration of OPN and that knockdown of VEGF increased the level of OPN again (*Figure 4H*).

These results demonstrated that the signal-connectors were effective tools for constructing regulatory loops and gene–gene networks.

## Signal-connectors specifically silence survival gene expression and inhibit cancer cell growth

To examine whether these devices could be used to identify cell state and to reprogram cellular behavior, we used the human telomerase reverse transcriptase (hTERT) promoter to drive the expression of ribozyme-flanked signal-connectors (*Gao and Zhao, 2014*) that silence survival genes, and chose bladder cancer cells as the target cells (*Figure 5A* and *Figure 5—figure supplement 1*). The hTERT promoter (hTERTp) is highly active in over 85% of human cancers, but inactive in most normal cells (*Takakura et al., 1999*). We therefore constructed device-ligand complexes to form a logical AND gate in which the activated hTERTp and the ligand must be combined to suppress the survival gene (*Figure 5B*). Signal-connectors suppressing the human c-Myc gene (*Sardi et al., 1998*)

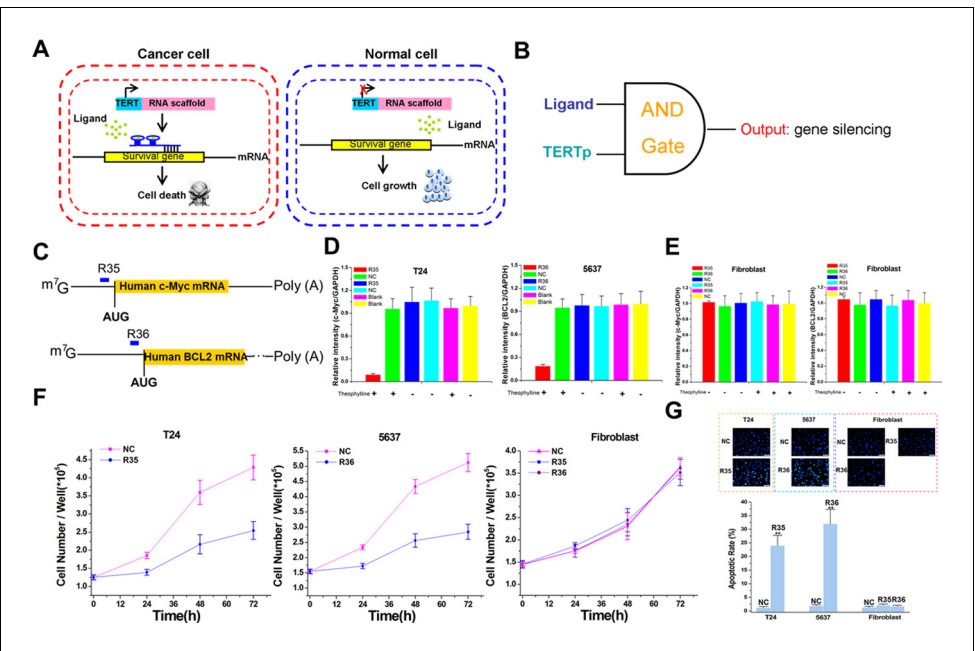

**Figure 5.** The signal-connectors specifically silence survival gene expression and inhibit cell growth in the targeted cancer cells. (**A**) Mechanisms of the signal-connectors designed to selectively kill cancer cells, which control cell survival in response to the presence of activated TERTp and ligand. (**B**) A schematic representation of the genetic AND gate. hTERT promoter and ligand (1000 μM theophylline) are the two inputs of the circuit. (**C**) Two different signal-connectors were designed to target the 5'-UTRs of human c-Myc mRNA and BCL2 mRNA. (**D and E**) Quantitative western blot analysis of targeted protein expression in bladder cancer cells (T24 and 5637) and normal fibroblast cells. NC, negative control vector with two repeated elements not having targets in the human genome. Blank, cells that were not transfected with vector. (**F**) Cell growth was measured by CCK-8 assay at different time intervals. ANOVA was used for the comparison of cell growth curves. Reported values are mean ± SD from three independent experiments.

DOI: https://doi.org/10.7554/eLife.31936.018

The following figure supplements are available for figure 5:

**Figure supplement 1.** Sequence and cleavage mechanism of the ribozymes.
DOI: https://doi.org/10.7554/eLife.31936.019

**Figure supplement 2.** Representative images of western blot analysis of c-Myc/BCL2 protein expression in bladder cancer cells transfected with the signal-connector or the controls.
DOI: https://doi.org/10.7554/eLife.31936.020

**Figure supplement 3.** Representative images of western blot analysis of c-Myc/BCL2 protein expression in Fibroblast transfected with the signal-connector or the controls.
DOI: https://doi.org/10.7554/eLife.31936.021

and the BCL2 gene (*Kunze et al., 2012*) were generated as before and stably transfected into either bladder cancer cells or normal dermal fibroblasts (*Figure 5C*, *Supplementary file 12* and *13*). In either bladder cancer cell line, the corresponding device was able to display significant decreases in gene expression in the presence of theophylline compared with that in the absence of ligand (*Figure 5D* and *Figure 5—figure supplement 2*). The devices did not lead to inhibitory effects in fibroblasts grown in the absence or presence of theophylline (*Figure 5E* and *Figure 5—figure supplement 3*). The growth curves of these cell lines also demonstrated that the circuit effectively inhibited proliferation of targeted bladder cancer cells without affecting the fibroblasts (*Figure 5F*). In addition, we then examined whether apoptosis of cancer cells can be induced by these devices. Bladder cancer cells treated with the signal connectors exhibited stronger blue fluorescence, revealing typical apoptotic characteristics. In contrast, the signal connectors had no such effects in the normal cells (*Figure 5G*). These results indicated that the AND gate circuit based on the signal-connectors could specifically suppress gene expression in the targeted cell lines.

## Redirection of oncogenic signaling to an anti-oncogenic pathway via the signal-connector

The successful application of signal-connector-mediated translational control in human cells opens the way toward a simultaneous ON/OFF multigene translational program in which some genes are activated and others are suppressed. We hypothesized that these devices should have the potential to redirect oncogenic pathway outputs and to control cancer cell fates through simultaneous activation and repression of endogenous genes.

NF-kB is an oncogenic signal that is known to be involved in the signaling pathways in cancer development. NF-kB controls cell proliferation by activating several downstream target genes such as cyclin D1, c-Fos and c-Jun (*Li et al., 2015*). We therefore sought to rewire NF-kB signaling from proliferation pathways to quiescence/death by using the signal-connector. We constructed four signal-connectors (*Supplementary file 14*, *15*, *16* and *17*) recognizing NF-kB (p65) (*ursterWurster and Maher, 2008*) to activate two tumor suppressors, Bax (R37) and p21 (R38), and to repress two tumor promoters, Bcl2 (R39) and c-myc (R40), in human bladder cancer T24 cells which normally expressed high levels of NF-kB signals (*Figure 6A*). In detail, two copies of eIF4G aptamer were inserted into the 3'end of NF-kB riboswitch to construct signal-connector activating Bax or p21, while two copies of NF-kB aptamer were linked with the 3'end of antisense RNA to construct the signal-connector suppressing Bcl2 or c-myc. The results of western blotting showed that the signal-connectors stably transfected in T24 cells could simultaneously enhanced the protein expression levels of Bax and p21 and decreased Bcl2 and c-myc (*Figure 6B* and *Figure 6—figure supplement 1*). Finally, we

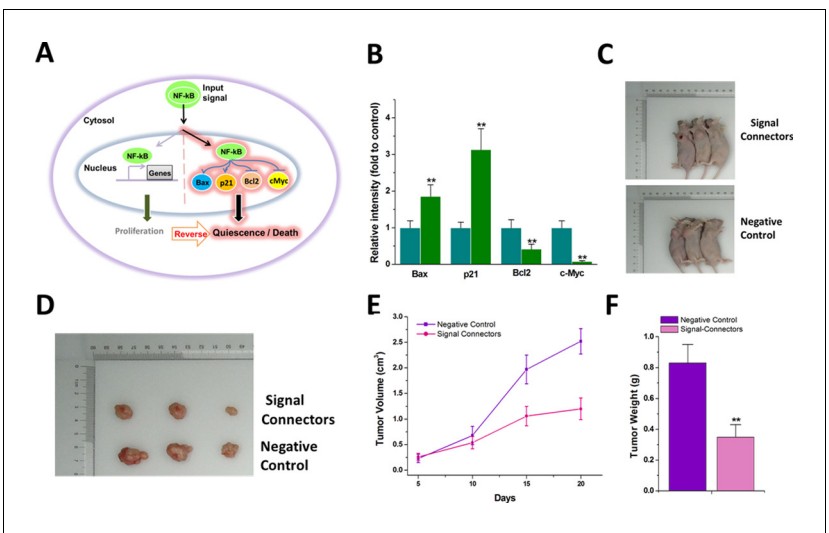

**Figure 6.** The signal-connector induces simultaneous activation and repression of cellular genes in response to oncogenic signal and redirects the oncogenic signaling to an anti-oncogenic pathway. (A) The oncogenic signal NF-kB was redirected to activate two tumor suppressors, Bax and p21, and to suppress two oncogenes, BCL2 and c-Myc, by the signal-connector in the cancer cells. (B) The relative expression levels of BAX, BCL2, c-Myc and p21 were determined in T24 cells by quantitative western blot at 48 hr after cell seeding. Reported values are mean ± SD from three independent experiments. (C and D) 20 days after injection, tumors formed in the signal-connector group were dramatically smaller relative to negative control. (E) The tumor volume was calculated once every 5 days after injection of T24 cells stably transfected with signal-connector or negative control. Bars indicate SD. (F) Tumor weights are shown as means of tumor weight ± SD. **p<0.01.
DOI: https://doi.org/10.7554/eLife.31936.022

The following figure supplements are available for figure 6:

**Figure supplement 1.** Representative images of western blot analysis of targeted protein expression in T24 cells transfected with the signal-connectors or the negative control.
DOI: https://doi.org/10.7554/eLife.31936.023

**Figure supplement 2.** Growth curves of T24 cell lines transfected with either the signal-connectors or the negative control vector.
DOI: https://doi.org/10.7554/eLife.31936.024

determined whether the signal-connectors could simultaneously inhibit tumor growth in vivo, since T24 cells stably transfected with these devices showed a slower in vitro growth rate (*Figure 6—figure supplement 2*). The cells were then inoculated into male nude mice. Twenty days after injection, we found that the tumors formed in the signal-connectors group were dramatically smaller than those in the negative control group (*Figure 6C–E*). In addition, the average tumor weight was markedly lower in the signal-connectors group compared to the negative control group at the end of the experiment (*Figure 6F*). These results indicate that signal-connectors could inhibit tumor growth in vivo by redirecting oncogenic signaling pathways.

## Discussion

In this study, we used antisense RNA to target the desired mRNA and the aptamer domain-ligand complex to repress translation of the target gene. Although it has been reported that an aptamer inserted in the 5'-UTR-mRNA can effectively repress translation of a synthetic gene (expressed in an exogenous vector) through the road-block mechanism, this is the first work to propose that aptamers could also repress endogenous genes of interest in a trans manner by linking with the antisense RNAs. It should also be noted that some natural non-coding RNAs (*Loh et al., 2009*; *Mellin et al., 2013*; *Price et al., 2015*; *Xu et al., 2015*) were found to inhibit their target gene through a similar mechanism, which suggests that the method used in this study is a universal approach for cellular RNAs to interact with genes of interest. More importantly, we showed that the aptamer tethering translation initiation factor could be used to enhance endogenous gene translation. In a similar case, this new mechanism was also proposed by one previous work which reported that the uchl1 gene lncRNA enhances the translation of its target mRNA via base-pairing and by recruiting additional ribosomes via the functional element (*Carrieri et al., 2012*). Therefore, this approach allows simultaneous activation and repression of different target genes, thus enabling robust reprogramming of cellular networks.

It is interesting that the signal-connectors mainly located in the cytoplasm. One possible explanation for this phenomenon is that the device binds the mRNA in the nucleus and remains bound to the mRNA during and after export. This observation is consistent with an earlier study which indicated that nuclear-localized sgRNA targeting mRNA can also be exported to the cytoplasm (*Nelles et al., 2016*).

In the applications of this methodology, we used two signal connectors targeting luciferase gene, but containing aptamers to different ligands, to construct circuits which are somewhat similar to computer logic gates. We also applied this methodology to couple unrelated signaling pathways and to connect oncogenic signals with an antioncogenic pathway. The signal connectors may provide an alternative approach to traditional cancer gene therapy which usually targets only one single gene.

In the construction of activator devices (with eIF4G), some basic thermodynamic/kinetics parameters, such as binding affinity and kinetics behaviors, still require quantitative studies in future works. The potential impact of the secondary structures in the targeted region of mRNA may also not be ignored. It seems that other small RNA regulators that are enzymatically amplified (such as shRNAs and miRNAs) exhibit much lower efficiencies than what we report here when expressed from the U6 promoter. We will compare efficacy of this new approach with that of the existing, simpler treatment method in future works.

Although conceptually simple, these devices may be included in the biology toolbox that allows for construction of novel signaling circuits and regulatory loops with predetermined properties and may enable development of strategies for treating disease networks. With regard to research applications of this methodology, one clear limitation is the fact that functional aptamers still usually need to be selected from random libraries. With the discovery and development of more and more aptamers, any signal not associated with gene regulation can be directed to inhibit/enhance translation of the targeted gene through the designed signal-connector which is very important to the signal transmission of parts of the gene circuits. Our results showed that these devices could be used to create genetic switches, logic gates, novel signaling and feedback loops, which led to practical applications such as detection of cancer cell state and inhibition of cancer cell survival. Signal-connectors that rewire multiple oncogenic signaling networks may provide an effective network-based strategy to increase the efficiency of current cancer treatment. In addition, mammalian cells

harboring digital logic gates can function as living bio-computers and open new avenues for artificial control of future gene- or cell-based therapies in a specific condition-dependent manner. Our novel technique will provide a useful platform for editing the common network structures and their signaling processes and will bring many applications in biology and medicine.

## Materials and methods

### Designing principle of RNA-based signal-connectors

We first analyzed the sequences of well-known RNA aptamers in mammalian cells, such as theophylline aptamer (*Jenison et al., 1994*), tetracycline aptamer (*Müller et al., 2006*), eIF4G aptamer (*Miyakawa et al., 2006*), β-catenin aptamer (*Culler et al., 2010*), VEGF aptamer (*Ng et al., 2006*), OPN aptamer (*Mi et al., 2009*) and NF-kB (p65) aptamer (urster et al., 2008). Then, we truncated and coupled them to the mRNA base pairing regions (antisense domains). Each mRNA base pairing region was perfectly complementary with the 5' –UTR or the coding region of the target mRNA. Next, the secondary structures of these recombinant RNAs were predicted by MFOLD program. The RNAs which showed exposed antisense domains and maintained the natural secondary structures of aptamers were selected and used in this study.

### Plasmids construction

The cDNA sequences for signal-connectors targeting Renilla luciferase/c-Myc/OPN/VEGF/ $BCL_2$/ Bax/p21 mRNA were synthesized and inserted into pGPU6/GFP/Puro vector at restriction site of Bam HI/Bbs I, respectively. Using similar approach, the cDNA sequences for ribozyme-flanked signal-connectors targeting c-Myc/$BCL_2$ mRNA were designed, synthesized, and inserted into hTERT-NEO-BAM vector at the restriction site of Sal I/BamH I, respectively. To construct plasmids pcDNA3.0-VEGF and pcDNA3.0-OPN, cDNA sequences expressing truncated forms of OPN/VEGF that lack the N-terminal signal peptide were inserted into pcDNA3.0 digested with BamHI/EcoRI, respectively. To construct plasmids shRNA-NC and shRNA-VEGF, the synthesized shRNA sequences were inserted into pGPU6/GFP/Neo digested with Bam HI/Bbs I, respectively.

### Cell lines and cell culture

T24, 5637, and HEK-293 cells were purchased from American Type Culture Collection (ATCC) by our laboratory and were grown in DMEM medium supplemented with 10% foetal bovine serum (Invitrogen, Carlsbad, CA) in the presence of 5% $CO_2$. Normal human primary fibroblasts derived from the epidermis were primary cultured in the same medium. T24, 5637, and HEK-293 cells have been previously authenticated by ATCC with STR profiling and no further authentication was done for these studies. Stable cell lines we re generated from these cell lines as described below. All cell lines used were validated as mycoplasma-free.

HEK-293 cells stably expressing Renilla luciferase were obtained by transfecting cells with pcDNA3/Rluc/Neo and selecting positive clones with G418. In details, stable selections were carried out in 6-well plates seeded with ~$2\times10^5$ HEK-293 cells per well, where 2 µg of the linearized plasmids were transfected using Lipofectamine 2000 Transfection Reagent (Invitrogen) according to the manufacturer's instructions. Cell monolayers were trypsinized 48 hr after transfection and transferred into T25 flasks or 100-mm-diameter culture dishes. A mixed population of stable transfectants was selected by growth in complete medium containing 500 µg of G418/ml. These multiclonal cell lines were expanded and then verified by luciferase reporter gene assay.

HEK-293 cells stably expressing either the signal-connector or the negative control were selected after transfection of the pGPU6/GFP/Puro vectors. HEK 293/T24 cells co-expressing multiple signal-connectors were constructed by stably transfecting a single pGPU6/GFP/ Puro vector which simultaneously generated these devices driven by a single U6 promoter. In details, cells were seeded in six-well plates and 2 µg of the linearized plasmids were transfected using Lipofectamine 2000 Transfection Reagent (Invitrogen) according to the manufacturer's instructions. After transfection, the cells were grown in the medium supplemented with puromycin at 4 µg/mL for approximately 14 days to select for a mixed population of stable cell lines. Then the multiclonal cells were verified by GFP expression. An inverted fluorescence microscope was used for direct observation of fluorescent cells in the culture plate.

## Luciferase reporter assay

HEK-293 cells stably expressing the Renilla luciferase reporter system were seeded in six-well plates ($5 \times 10^5$/ well). Forty-eight hours after transfection, the medium was removed and cells were lysed in 500 µl of lysate buffer (Analytical Luminescence Laboratories). Renilla luciferase activity was measured by the Renilla Luciferase Reporter Assay System (Promega, Madison, WI) according to manufacturer's instructions. Renilla luciferase activities were corrected for variation in protein concentrations of the cell extracts (Bio-Rad). The assays were performed in duplicate and the experiments were repeated three times.

## Nuclear and cytoplasmic RNA fractionation analysis

Nuclear and cytoplasmic RNA were isolated using the Cytoplasmic and Nuclear RNA Purification Kit (Norgen, Belmont, CA) according to the provider's instructions.

## In vitro translation reaction

Purified ligand (theophylline or eIF4G) was incubated with 1 µg signal connector and 1 µg Renilla luciferase mRNA and then the mixture was further incubated with the components of the Thermo Scientific 1-Step Human Coupled IVT Kit (Waltham, MA, USA) according to the manufacturer's instructions. The activity of in vitro translated Renilla luciferase was calculated as described above.

## ELISA assay of VEGF/OPN concentration

HEK293 cells were stably transfected with signal-connectors or the control. The concentration of VEGFA/OPN protein was then measured by ELISA assay, which was employed according to the manufacturer's instructions. Briefly, $10^6$/sample cells were harvested and resuspended in 200 µl of lysis buffer. The supernatants of lysates were collected through centrifugation and used for the following procedures. The OD values were then measured by a microplate reader (Bio-Rad, Hercules, CA) and converted to protein concentrations using standard calibration curves.

## Western blot analysis

Cells were washed in PBS and lysed in RIPA buffer (50 mM Tris-HCl pH 7.2, 150 mM NaCl, 1% NP40, 0.1% SDS, 0.5% DOC, 1 mM PMSF, 25 mM $MgCl_2$, and supplemented with a phosphatase inhibitor cocktail). The protein concentration was determined using the BCA protein assay. Equal amounts of whole protein extract were electrophoresed onto SDS–polyacrylamide gels and then transferred to PVDF membranes (Millipore, Billerica, MA). Samples were blocked in 5% dry milk and incubated over-night with the primary antibodies (Abcam, Cambridge, MA). Then, the samples was incubated with horseradish peroxidase–conjugated secondary antibody (Amersham, Piscataway,NJ) and immunoblots were developed with Super Signal chemiluminescence reagents (Pierce Chemical Co.). The protein bands were quantified using Image J analysis software (National Institutes of Health, USA). Histograms were generated by normalizing the amount of each protein to the GAPDH level detected in the same extracted sample. Each experiment was repeated three times.

## Cell proliferation assay

Cell numbers were calculated by treating the cells with 0.25% trypsin (15 min, 37°C), followed by analysis on an electronic cell counter (Beckman Coulter) at 0, 24, 48 and 72 hr. The assay was repeated at least three times independently.

## Cell apoptosis assay

The Hoechst 33258 staining kit (Life, Eugene, OR) was used to observe the apoptotic cells induced by signal-connectors. Briefly, the treated cells were fixed in 4% paraformaldehyde for 10 min and washed twice in PBS. Then, the cells were stained with 0.5 ml of Hoechst 33258 staining for 5 min and photos were taken under a fluorescence microscope at a wavelength of 350 nm. Each assay was repeated three times.

## RNA extraction and real-time quantitative PCR

Total RNA was isolated from cells by using TRIzol (Invitrogen, Carlsbad, CA) according to the suggested protocol. The cDNA strand was synthesized from total RNA with RevertAidTM First Strand

cDNA Synthesis Kit (Fermentas, Hanover, MD) in a 25 µl volume. Real time quantitative PCR was performed with the All-in-OneTM qPCR Mix (GeneCopoiea Inc, Rockville, MD) in a 20 µl reaction volume on an ABI PRISM 7000 Fluorescent Quantitative PCR System (Applied Biosystems, Foster City, CA). The PCR cycling parameters were: 95℃ for 15 min, followed by 40 cycles of 94℃ for 15 s, 55℃ for 30 s and 72℃ for 30 s. Relative expression fold changes were determined by the $2^{-\Delta\Delta Ct}$ method.

## Tumor formation assay in nude mouse model

All experiments involving animals were approved by Institutional Review Board. Four- week-old female BALB/c nude mice were obtained from Animals Center of the Academy of Sciences. In detail, $10^7$ T24 cells stably expressing signal-connector or negative control were suspended in 100 µl PBS and injected subcutaneously into left or right armpits of three 4-week-old female BALB/c nude mice. Tumor growth was examined every 5 days, and tumor volumes were also calculated using the formula: $0.5 \times length \times width^2$. 20 days after injection, mice were euthanized, and the subcutaneous weight of each tumor was measured.

## Statistical analyses

No statistical methods were used to pre-determine sample size. The investigators were blinded to allocation during experiments and outcome assessment. Statistical analysis was conducted using Student's t-test or ANOVA and $p<0.05$ was considered statistically significant. All statistical tests were performed by using SPSS version 17.0 software (SPSS, Chicago, IL).

## Acknowledgements

We are indebted to the donors whose names were not included in the author list, but who participated in this program. This work was supported by the National Key Basic Research Program of China (973 Program) (2014CB745201 to ZC), National Natural Science Foundation of China (81402103; 81773257 to YL), and Special Support Funds of Shenzhen for Introduced High-Level Medical Team to ZC.

## Additional information

### Funding

| Funder | Grant reference number | Author |
|---|---|---|
| National Natural Science Foundation of China | 81773257 | Yuchen Liu |
| National Natural Science Foundation of China | 81402103 | Yuchen Liu |
| National Key Basic Research Program of China | 2014CB745201 | Zhiming Cai |
| Special Support Funds of Shenzhen for Introduced High-Level Medical Team | | Zhiming Cai |

The funders had no role in study design, data collection and interpretation, or the decision to submit the work for publication.

### Author contributions

Yuchen Liu, Conceptualization, Data curation, Formal analysis, Supervision, Funding acquisition, Validation, Investigation, Methodology, Writing—original draft, Project administration, Writing—review and editing; Jianfa Li, Investigation, Methodology, Writing—review and editing; Zhicong Chen, Writing—original draft, Project administration, Writing—review and editing; Weiren Huang, Conceptualization, Resources, Funding acquisition, Methodology; Zhiming Cai, Resources, Supervision, Funding acquisition, Methodology, Project administration, Writing—review and editing

**Author ORCIDs**

Yuchen Liu ⓘ http://orcid.org/0000-0002-6517-0022

**Ethics**

Animal experimentation: This study was performed in strict accordance with the recommendations in the Guide for the Care and Use of Laboratory Animals of the National Institutes of Health. All mice were housed and handled in accordance with protocols approved by the Committee on the Use of Live Animals in Teaching and Research of Shenzhen University. To minimize suffering, all surgeries were performed under anesthesia.

**Decision letter and Author response**

Decision letter https://doi.org/10.7554/eLife.31936.044
Author response https://doi.org/10.7554/eLife.31936.045

# Additional files

**Supplementary files**

• Supplementary file 1. cDNA sequences of the theophylline-induced signal-connectors targeting and suppressing Renilla luciferase mRNA translation. Each of these sequences consists of a complementary sequence, two copies of theophylline aptamers, and a linker sequence.
DOI: https://doi.org/10.7554/eLife.31936.025

• Supplementary file 2. cDNA sequences of the tetracycline-induced signal-connectors targeting and suppressing Renilla luciferase mRNA translation. Each of these sequences consists of a complementary sequence, two copies of tetracycline aptamers, and a linker sequence.
DOI: https://doi.org/10.7554/eLife.31936.026

• Supplementary file 3. cDNA sequences of the tetracycline-induced signal-connectors targeting and suppressing VEGF mRNA translation. Each of these sequences consists of a complementary sequence, two copies of tetracycline aptamers, and a linker sequence.
DOI: https://doi.org/10.7554/eLife.31936.027

• Supplementary file 4. The cDNA sequence of the signal-connector targeting and enhancing Renilla luciferase mRNA translation. The sequence consists of a complementary sequence, two copies of eIF4G aptamers, and a linker sequence.
DOI: https://doi.org/10.7554/eLife.31936.028

• Supplementary file 5. cDNA sequences of theophylline-induced signal-connectors targeting and enhancing Renilla luciferase mRNA translation. Each of these sequences consists of a complementary sequence, one copy of theophylline riboswitch, two copies of eIF4G aptamers and two linker sequences.
DOI: https://doi.org/10.7554/eLife.31936.029

• Supplementary file 6. cDNA sequences of tetracycline-induced signal-connectors targeting and enhancing Renilla luciferase mRNA translation. Each of these sequences consists of a complementary sequence, one copy of tetracycline riboswitch, two copies of eIF4G aptamers and two linker sequences.
DOI: https://doi.org/10.7554/eLife.31936.030

• Supplementary file 7. The cDNA sequence of β-catenin-induced signal-connector targeting and suppressing c-Myc mRNA translation. The sequence consists of a complementary sequence, two copies of β-catenin aptamers, and a linker sequence.
DOI: https://doi.org/10.7554/eLife.31936.031

• Supplementary file 8. The cDNA sequence of β-catenin-induced signal-connector targeting and enhancing Renilla luciferase mRNA translation. The sequence consists of a complementary sequence, one copy of β-catenin riboswitch, two copies of eIF4G aptamers and two linker sequences.
DOI: https://doi.org/10.7554/eLife.31936.032

• Supplementary file 9. The cDNA sequence of VEGF-induced signal-connector targeting and enhancing OPN mRNA translation. The sequence consists of a complementary sequence, one copy of VEGF riboswitch, two copies of eIF4G aptamers and two linker sequences.

DOI: https://doi.org/10.7554/eLife.31936.033

• Supplementary file 10. The cDNA sequence of OPN-induced signal-connector targeting and enhancing VEGF mRNA translation. The sequence consists of a complementary sequence, one copy of OPN riboswitch, two copies of eIF4G aptamers and two linker sequences.
DOI: https://doi.org/10.7554/eLife.31936.034

• Supplementary file 11. The cDNA sequence of VEGF-induced signal-connector targeting and suppressing OPN mRNA translation. The sequence consists of a complementary sequence, two copies of VEGF aptamers and one linker sequence.
DOI: https://doi.org/10.7554/eLife.31936.035

• Supplementary file 12. The cDNA sequence of theophylline-induced signal-connector targeting and suppressing c-Myc mRNA translation. The sequence consists of a complementary sequence, two copies of theophylline aptamers and one linker sequence.
DOI: https://doi.org/10.7554/eLife.31936.036

• Supplementary file 13. The cDNA sequence of theophylline-induced signal-connector targeting and suppressing BCL2 mRNA translation. The sequence consists of a complementary sequence, two copies of theophylline aptamers and one linker sequence.
DOI: https://doi.org/10.7554/eLife.31936.037

• Supplementary file 14. The cDNA sequence of NF-kB-induced signal-connector targeting and enhancing Bax mRNA translation. The sequence consists of a complementary sequence, one copy of NF-kB riboswitch, two copies of eIF4G aptamers and two linker sequences.
DOI: https://doi.org/10.7554/eLife.31936.038

• Supplementary file 15. The cDNA sequence of NF-kB-induced signal-connector targeting and enhancing p21 mRNA translation. The sequence consists of a complementary sequence, one copy of NF-kB riboswitch, two copies of eIF4G aptamers and two linker sequences.
DOI: https://doi.org/10.7554/eLife.31936.039

• Supplementary file 16. The cDNA sequence of NF-kB-induced signal-connector targeting and suppressing BCL2 mRNA translation. The sequence consists of a complementary sequence, two copies of NF-kB aptamers and one linker sequence.
DOI: https://doi.org/10.7554/eLife.31936.040

• Supplementary file 17. The cDNA sequence of NF-kB-induced signal-connector targeting and suppressing c-Myc mRNA translation. The sequence consists of a complementary sequence, two copies of NF-kB aptamers and one linker sequence.
DOI: https://doi.org/10.7554/eLife.31936.041

• Transparent reporting form
DOI: https://doi.org/10.7554/eLife.31936.042

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
