## [Decision Letter]

Thank you for submitting your article "Synthesizing artificial devices that redirect cellular information at will" for consideration by *eLife*. Your article has been favorably evaluated by Naama Barkai (Senior Editor) and two reviewers, one of whom is a member of our Board of Reviewing Editors. The reviewers have opted to remain anonymous.

The reviewers have discussed the reviews with one another and the Reviewing Editor has drafted this decision to help you prepare a revised submission.

Summary:

The article describes a novel RNA-based trans-regulatory system to control gene expression in mammalian cells. The regulators exhibit good dynamic range and they can be coupled to endogenous signals. Complex regulation patterns corresponding to various 2-input logic combinations were also shown. Overall the study describes a novel, well-performing and potentially useful framework for combinatorial control of gene expression using RNA regulators in trans.

Essential revisions:

1) The bulk of expression is reported based on luciferase assay. Moreover, only one luciferase has been knocked in without internal reference. The signal seems to have been measured following cell lysis. Was the number of cells controlled for by cell counting? It is very possible that the difference in signal resulted from difference in cell numbers between wells, for example due to difference in cell growth rate. The authors should explicitly state in much more detail how the luciferase-based data were obtained.

2) The measurements are always shown relative to control. It would be helpful to see the different data in absolute units, e.g. expression per cell.

3) The authors should describe in much more detail the rationale for their short RNA regulator design. Were the reported sequences selected from a large library or rationally designed and worked as expected in the first trial?

4) The cells used in the experiment are all stable cell lines. This is a welcome approach, however the selection of clones is unclear. Were uniform criteria applied to all clones? Were the different clones screened functionally prior to reporting? How robust is the effect across clones? What is the integration copy number? In summary, as much detail as possible should be provided regarding the clones, including data sheets for each clone and their characterisation such as GFP expression (as FACS plot) and baseline luciferase activity in absolute units per cell. Ideally, genotyping would also be helpful.

5) The tumor in vivo experiments are not entirely convincing. It seems as if the cells modified with the NFkB sensor had slower growth. These cells should be characterized in vitro for cell growth rates, as the reviewers suspect that the differential tumor progression simply reflects the different growth rate in vitro. The reviewers do not underestimate the success of slowing down cell growth, but the data need to be shown for completeness.

---

## [Author Response]

Essential revisions:1) The bulk of expression is reported based on luciferase assay. Moreover, only one luciferase has been knocked in without internal reference. The signal seems to have been measured following cell lysis. Was the number of cells controlled for by cell counting? It is very possible that the difference in signal resulted from difference in cell numbers between wells, for example due to difference in cell growth rate. The authors should explicitly state in much more detail how the luciferase-based data were obtained.

Thanks for your valuable comments. Forty-eight hours after transfection, the medium was removed and cells were lysed in 500μl of lysate buffer (Analytical Luminescence Laboratories). *Renilla* luciferase activity was measured by the *Renilla* Luciferase Reporter Assay System (Promega, Madison, WI, USA) according to manufacturer’s instructions. *Renilla* luciferase activities were corrected for variation in protein concentrations of the cell extracts (Bio-Rad).The assays were performed in duplicate and the experiments were repeated three times. We assumed that same amount of total protein indicated same total cell numbers.

The above details have also been included in the Materials and methods Section.

2) The measurements are always shown relative to control. It would be helpful to see the different data in absolute units, e.g. expression per cell.

Thanks for your valuable comments. As requested, we used “luciferase units /μg protein” to replace “relative luciferase activity (%)”. Please see Figure 2, Figure 3 and Figure 4 for more details.

3) The authors should describe in much more detail the rationale for their short RNA regulator design. Were the reported sequences selected from a large library or rationally designed and worked as expected in the first trial?

Thanks for your valuable comments. The reported RNA sequences have been rationally designed based on the reported RNA aptamers. Most of these RNA devices worked as expected in the first trial. We have included a new paragraph in the Materials and methods section to further introduce the design principle of RNA-based signal-connectors: “We first analyzed the sequences of well-known RNA aptamers in mammalian cells, such as theophylline aptamer (Jenison et al., 1994), tetracycline aptamer (Muller et al., 2006), eIF4G aptamer (Miyakawa et al., 2006), β-catenin aptamer (Culler et al., 2010),VEGF aptamer (Ng et al., 2006), OPN aptamer (Mi et al., 2009) and NF-κB (p65) aptamer (urster et al., 2008). […] The RNAs which showed exposed antisense domains and maintained the natural secondary structures of aptamers were selected and used in this study”.

4) The cells used in the experiment are all stable cell lines. This is a welcome approach, however the selection of clones is unclear. Were uniform criteria applied to all clones? Were the different clones screened functionally prior to reporting? How robust is the effect across clones? What is the integration copy number? In summary, as much detail as possible should be provided regarding the clones, including data sheets for each clone and their characterisation such as GFP expression (as FACS plot) and baseline luciferase activity in absolute units per cell. Ideally, genotyping would also be helpful.

Thanks for your valuable comments. We are sorry for not clearly showing the details of the selection of stable cell lines. We used multiclonal cell lines (without isolation of specific clones) in this study. Because stable integration is often accompanied by the inactivation of an endogenous gene from the host, the use of mixed stable cell lines in experiments can help to obtain more accurate data.

The details of the selection of stable cell lines have been included in the Materials and methods section:

“HEK-293 cells stably expressing *Renilla* luciferase were obtained by transfecting cells with pcDNA3/Rluc/Neo and selecting positive clones with G418. […] An inverted fluorescence microscope was used for direct observation of fluorescent cells in the culture plate.”

5) The tumor in vivo experiments are not entirely convincing. It seems as if the cells modified with the NFkB sensor had slower growth. These cells should be characterized in vitro for cell growth rates, as the reviewers suspect that the differential tumor progression simply reflects the different growth rate in vitro. The reviewers do not underestimate the success of slowing down cell growth, but the data need to be shown for completeness.

Thanks for your valuable comments. We do agree with the view of the reviewers. In fact, we have calculated the in vitro growth rates of these cells in our preliminary results. We found that T24 cells stably transfected with these devices showed a slower in vitro growth rate. We have added these data into the manuscript (Figure. 6—figure supplement 2).